# *CDHR1*-Related Cone–Rod Dystrophy: Clinical Characteristics, Imaging Findings, and Genetic Test Results—A Case Report

**DOI:** 10.3390/medicina59020399

**Published:** 2023-02-17

**Authors:** Małgorzata Sobolewska, Marta Świerczyńska, Mariola Dorecka, Dorota Wyględowska-Promieńska, Maciej R. Krawczyński, Ewa Mrukwa-Kominek

**Affiliations:** 1Department of Ophthalmology, Kornel Gibiński University Clinical Center, Medical University of Silesia, 40-514 Katowice, Poland; 2Department of Ophthalmology, Faculty of Medical Sciences in Katowice, Medical University of Silesia, 40-514 Katowice, Poland; 3Department of Medical Genetics, Poznan University of Medical Sciences, 60-406 Poznan, Poland; 4Centers for Medical Genetics GENESIS, 60-406 Poznan, Poland

**Keywords:** cone–rod dystrophy, CRD, *CDHR1* gene, ffERG

## Abstract

*Background:* Cone–rod dystrophies (CRDs) are a heterogeneous group of inherited retinal diseases (IRDs) characterized by cone photoreceptor loss, that is followed by subsequent rod photoreceptor impairment. *Case presentation:* A 49-year-old man complaining of diminution of vision in both eyes (OU) was referred to our outpatient clinic. He reported visual loss for 5 years, but it was most progressive during the last few months. The best-corrected visual acuity (BCVA) at presentation was 0.4 in the right eye (RE) and 1.0 in the left eye (LE). Fundus fluorescein angiography (FFA) revealed granular hyperfluorescence in the macula and concomitant areas of capillary atrophy. Flash full-field electroretinography (ffERG) showed lowering of a and b waves as well as prolonged peak time in light-adapted conditions. However, outcomes of dark-adapted ERGs were within normal limits. Based on the constellation of clinical, angiographic, and electrophysiological tests findings, a diagnosis of IRD was suspected. Genetic testing showed a homozygous, pathogenic c.783G>A mutation in the cadherin-related family member 1 (*CDHR1*) gene, which confirmed CRD type 15 (CRD15). *Conclusions:* We demonstrate the clinical characteristics, retinal imaging outcomes, and genetic test results of a patient with CRD15. Our case contributes to expanding our knowledge of the clinical involvement of the pathogenic mutation c.783G>A in *CDHR1* variants.

## 1. Introduction

Inherited retinal diseases (IRDs) comprise a heterogenous group, classically divided into stationary (with congenital or early-infantile onset) and progressive (usually with later onsets) and includes cone dystrophy (CD), cone–rod dystrophy (CRD), and rod–cone dystrophy (RCD, also known as retinitis pigmentosa (RP)) [1,2]. CRDs may be associated with Bardet Biedl syndrome, Spinocerebellar Ataxia Type 7, as well as ectodermal diseases (amelogenesis imperfecta, hypotrichosis with juvenile macular dystrophy, dysmorphic syndromes, metabolic dysfunctions) [3].

In CRDs, the loss of cone and rod photoreceptor cells leads to color vision disturbance, light intolerance, night blindness, and decreased visual acuity (VA) and sensitivity in the central visual field, later followed by the worsening of peripheral vision [3]. The first symptoms usually begin in the first or second decade of life. The prevalence of CRDs is estimated at 1:30,000–40,000 [4]. An autosomal recessive (AR) pattern of inheritance is the most common (60–70%), followed by autosomal dominant (AD) (20–30%) and X-linked (5%) [5,6]. Mutations in at least 34 genes are known to cause CDs/CRDs and many of these genes encode proteins involved in photoreceptor structure or the phototransduction cascade [7]. *ABCA4* is a primary causative gene that is responsible for 30–60% of AR CRDs cases in European populations [8].

Here, we present a 49-year-old patient with CRD related to a homozygous, pathogenic c.783G>A mutation in the cadherin-related family member 1 (*CDHR1*) gene. However, his response in dark-adapted (DA) conditions during electrophysiological examination was normal. Our report contributes to expanding our knowledge of the clinical involvement of the pathogenic c.783G>A mutation in *CDHR1* gene variants.

## 2. Case Report

A 49-year-old Caucasian male patient was referred to the outpatient clinic due to deterioration of vision in both eyes (OU) for last few months. He had been reporting problems with clarity of vision for 5 years, but in recent months he noticed a significant decrease in VA. There was no previous history of eye disease, trauma, or family history of inherited diseases. At presentation, the best-corrected visual acuity (BCVA) was 0.4 in the right eye (RE) and 1.0 in the left eye (LE), whereas Snellen test was 3.0 and 1.5, respectively. The Ishihara test was within normal limits in OU. The patient did not complain about night blindness. The anterior segment examination did not disclose any significant abnormality. Fundus examination revealed an atrophic area around the optic discs, macular degenerations, and retinal thinning in OU (Figure 1A,B).

Enhanced depth imaging optical coherence tomography (EDI-OCT) pictures revealed foveal thinning with distortion and loss of the outer nuclear layer (ONL), external limiting membrane (ELM), ellipsoid zone (EZ), and retinal pigment epithelium (RPE) bilaterally. The central foveal thickness was 175 μm for the RE and 170 μm for the LE, whereas the choroid thickness (CT) was 275 μm and 276 μm, respectively (Figure 2A,B). Static perimetry demonstrated a deepening central scotoma with saving peripheral vision in OU (Figure 3A,B).

Fundus autofluorescence (FAF) showed a centrally located area with irregular borders showing hypoautofluorescence because of RPE atrophy with a surrounding rim of hyperautofluorescence consistent with RPE damage. There was an area of hypoautofluorescence around the optic discs at the site of atrophy as well. (Figure 1C,D). The above changes correlated with the visual field. Fundus fluorescein angiography (FFA) revealed granular hyperfluorescence in the macula and concomitant areas of fenestrated loss with visible choriocapillaris. The contrast flow through the main vessels did not show any obvious abnormalities (Figure 4A–D).

Flash full-field electroretinography (ffERG) were performed using the RETeval portable unit (LKC, USA) and sensor strip electrodes, in accordance with the standards of the International Society for Clinical Electrophysiology of Vision [9]. There was lowering of a and b waves in light-adapted (LA) 3 ERG and LA 30 Hz ERG as well as prolonged peak times of a and b waves under photopic adaptation conditions (significantly worse in the RE). However, dark-adapted (DA) 0.01 ERG and DA3 were normal (Figure 5A,B).

Based on the constellation of clinical symptoms and diagnostic test results, a diagnosis of IRD was suspected. The patient underwent next-generation sequencing (NGS) of a panel of 317 macular, cone/cone–rod dystrophy, retinitis pigmentosa and other IRD-related genes. Enrichment was conducted using the Roche/NibleGen (Madison, WI, USA) sequence capture technology, and NGS was carried out on an Illumina HiSeq 1500 system (San Diego, CA, USA). Verification of identified variant and segregation analyses were carried out by PCR amplification of the corresponding exon, followed by Sanger sequencing. Genetic testing showed a homozygous mutation in *CDHR1* (NM_033100.4:c.783G>A(;)(783G>A). This splice region variant is classified in the Human Gene Mutation Database as pathogenic. Recessive cone–rod dystrophy type 15 (CRD/CORD 15, OMIM #613660) was diagnosed (Figure 6). No other putative pathogenic variants were reported from the genetic panel. In the next two-year follow-up, the BCVA lowered to 0.2 in the RE and 0.4 in the LE.

## 3. Discussion

Among the many symptoms over the course of CDs/CRDs, reduced VA is one of the earliest and usually appears in the first decade of life. Color vision disturbances found in CRDs show up as abnormalities in the perception of all three color axes, which correspond to the three subtypes of opsins. However, some studies provide evidence that protan and tritan defects are more frequently observed [10,11]. CDs are distinguished from CRDs by the absence of early nyctalopia, which appears in CRDs as a result of concomitant rod degeneration. It should be noted that it also emerges in the course of CDs, although in more advanced stages of the disease [12]. There is also considerable overlap of the phenotypes and the causative genes between CDs/CRDs and RP [13]. However, in contrast to CRDs, in RP, rod degeneration precedes the loss of cones, and the primary symptom is night blindness, which could be isolated for several years. Moreover, there is loss of peripheral vision and pigment deposits localized to the periphery of the retina [3].

Fundoscopic features of CRDs commonly include bilateral and relatively symmetric RPE distortion and/or atrophy progressing over time, whereas characteristic clinical finding is a bull’s-eye maculopathy. Peripheral RPE atrophy and bone spicule pigmentation have been observed in advanced stages of CRDs [7]. In FAF, there is reduced autofluorescence that occurs due to either the reduction or absence of lipofuscin as well as the presence of obstructing material prior to RPE and photoreceptor degeneration. The ring of increased FAF that is seen in CDs/CRDs is comparable in appearance to that in RP. However, unlike RP, the retina inside the ring has impaired function and the ring increases in size with disease progression [14].

Significant finding in OCT images are irregularities in the paracentral EZ and RPE in the early stages of the disease with relative preservation of the fovea, and the ONL appears thin in the affected areas. Subsequently, there is a loss of photoreceptors and the RPE layer, leading to a direct location of the inner retinal layers over the Bruch’s membrane [15]. While some studies reported a significant reduction of CT in CD [16], Sabbaghi et al. [17] using EDI-OCT demonstrated that mean CT was significantly lower only among patients with RP and Usher syndrome compared to healthy control group, and patients with Stargardt disease and CRDs; this was also revealed in our case report.

For CRDs, the ERG discloses impairment of both cone and rod function with a predominant reduction of photopic over scotopic responses. The ffERG is a crucial examination, especially when patients are asymptomatic and funduscopic appearance is normal. It is important to ascertain the diagnosis by repeating the test one or two years after it has been made [3].

In Stargardt disease, the peripheral retina usually remains unaffected and due to the presence of yellow flecks (sometimes covering the entire fundus), hyperfluorescent macular lesion (bull’s eye), and dark choroid in FFA imaging, it is fairly easy to diagnose. Interestingly, Stargardt disease is caused by a mutation in *ABCA4*, which also contributes to the onset of CRDs. In these cases, in the early stages of CRDs the fundus appearance may be comparable with that of Stargardt disease, but within a decade, peripheral retinal changes and associated symptoms will appear [3].

For cases where CDR is caused by a homozygous c.783G>A mutation in *CDHR1*, as shown by Charbel Issa et al. [15], the phenotypic picture of the disease is remarkably similar to that of central areolar choroidal dystrophy (CACD). Furthermore, symptoms of *CDHR1*-associated maculopathy do not appear before the fourth decade of life and due to the emerging atrophic changes associated with advanced disease, it is often misdiagnosed as age-related macular degeneration (AMD). It seems reasonable to consider *CDHR1*-associated retinopathy as a differential diagnosis, especially in cases of atypical AMD and familial occurrence with a pattern of recessive inheritance. In addition, the extended maculopathies that occur in AMD may be difficult to distinguish from end-stage CRDs or RP; in those cases, ffERG examination is crucial to differentiate. Additionally, multifocal ERG (mfERG) can be used to differentiate between local and diffuse cone dysfunction.

According to the reported symptoms and test results, our patient was diagnosed with IRD. He did not report any night visual disturbance and the Flash ffERG test demonstrated a normal response in DA conditions, which supported the diagnosis of CD rather than CRD. However, the genetic testing revealed a homozygous pathogenic mutation (c.783G>A) in *CDHR1*. CDHR1 is an adhesion molecule located primarily at the junction of the inner and outer segments of rod and cone photoreceptors and plays a crucial role in the maintenance of photoreceptor structure and integrity [6]. The phenotypic appearance of the retina depends on the severity of the mutation: homozygosity for the c.783G>A mutation results in disease essentially confined to the macula, whereas biallelic truncating mutations lead to both cone and rod dystrophies involving the entire retina along with severe loss of its function [15].

However, it is postulated that the c.783 G>A variant represents a hypomorphic allele associated with reduced levels of *CDHR1* transcripts (and eventually CDHR1 protein) that may be unmasked as being deleterious if occurring in a homozygous state or in a heterozygous state with another pathogenic allele [6]. The 783G>A variant is predominantly described in Caucasians and according to a study by Bessette et al. [18], we can conclude that the prevalence might be higher among the Polish population.

When the diagnosis of CRD is made, patients should be informed and a familial survey should be recommended. A precise phenotypic diagnosis is always mandatory and is particularly useful in the absence of familial history or in sporadic cases. Prenatal diagnosis can be performed in families in which the responsible gene has been identified [3]. Until now, no treatment has been proven effective in retarding the progression of CRDs. However, animal models of the disease with mutations in *GUCA1A*, *PRPH2*, *ABCA4*, and *RPGR* has demonstrated a significant improvement in photoreceptor survival rate after gene-based therapy introduction [19,20,21,22]. *CDHR1*-associated retinopathy therefore remains a disease meriting attention and further research concerning therapies based on gene editing. Moreover, there seems to be a significant time interval between the appearance of the first symptoms and severe vision loss, creating an adequate therapeutic window [15].

## 4. Conclusions

In summary, CRDs belong to the group of IRDs, in which deprivation of rods and cones leads to progressive visual loss. Advances in the field of visual function research and retinal imaging has improved our knowledge concerning the relationship between genotype and phenotype. Nevertheless, in atypical cases, such as our patient with a normal DA response, the genetic diagnostics have a crucial role in establishing an accurate diagnosis.

## Figures and Tables

**Figure 1 medicina-59-00399-f001:**
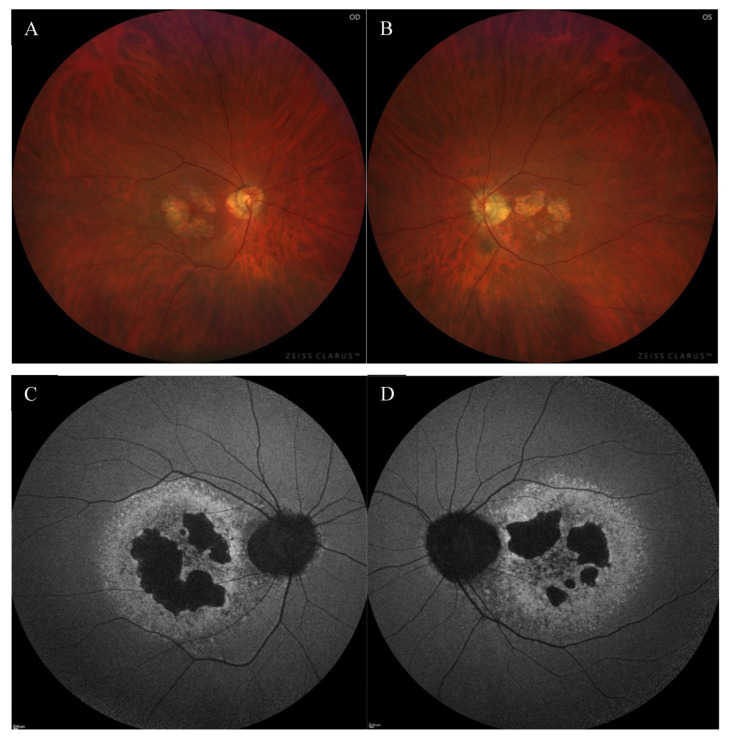
The color fundus images of the right (**A**) and the left (**B**) eye showed macular degenerations, retinal thinning, as well as an atrophic area around the optic disc. FAF of both eyes (**C**,**D**) disclosed hypoautofluorescence in the macula with a surrounding rim of hyperautofluorescence, as well as hypoautofluorescence around the optic disc.

**Figure 2 medicina-59-00399-f002:**
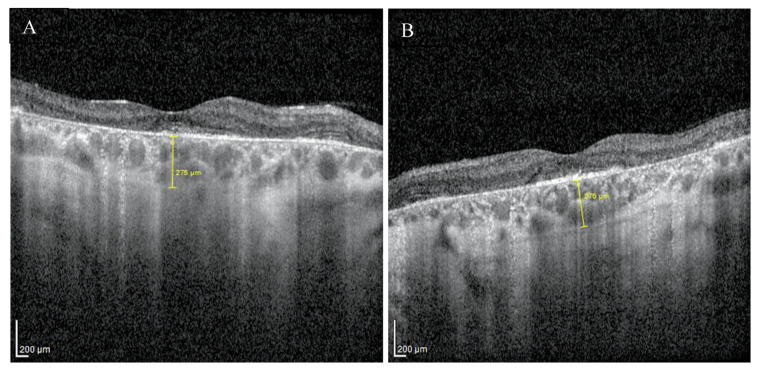
EDI-OCT pictures of the right (**A**) and the left (**B**) macula revealed atrophy of outer layers with normal choroid thickness.

**Figure 3 medicina-59-00399-f003:**
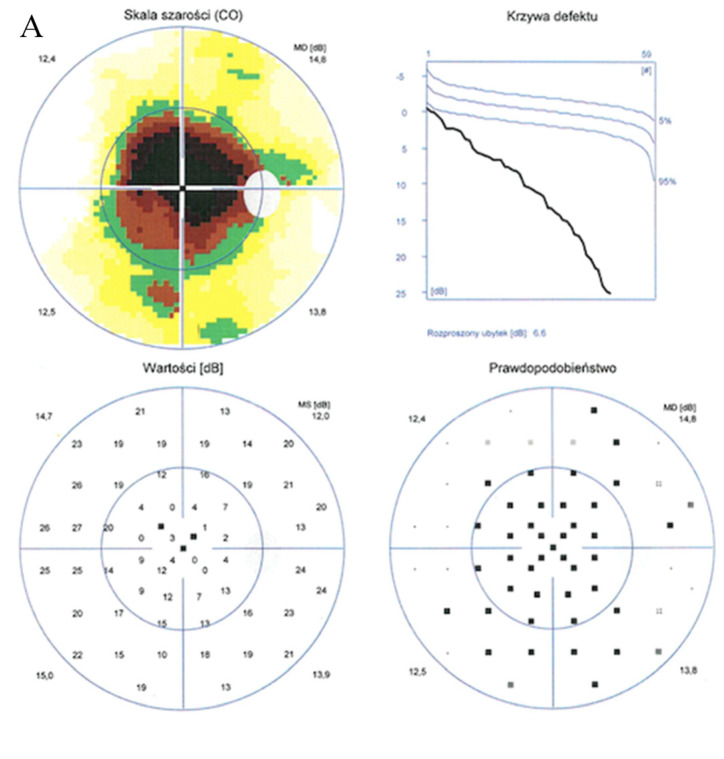
Static perimetry showed a central scotoma with saving peripheral vision in both eyes. The right eye (**A**) MS (mean sensitivity): 13.7 dB; MD (mean deviation): 13.6; sLV (standard loss variance): 9.8. The left eye (**B**) MS: 13.6 dB; MD: 13.8; sLV: 10.8.

**Figure 4 medicina-59-00399-f004:**
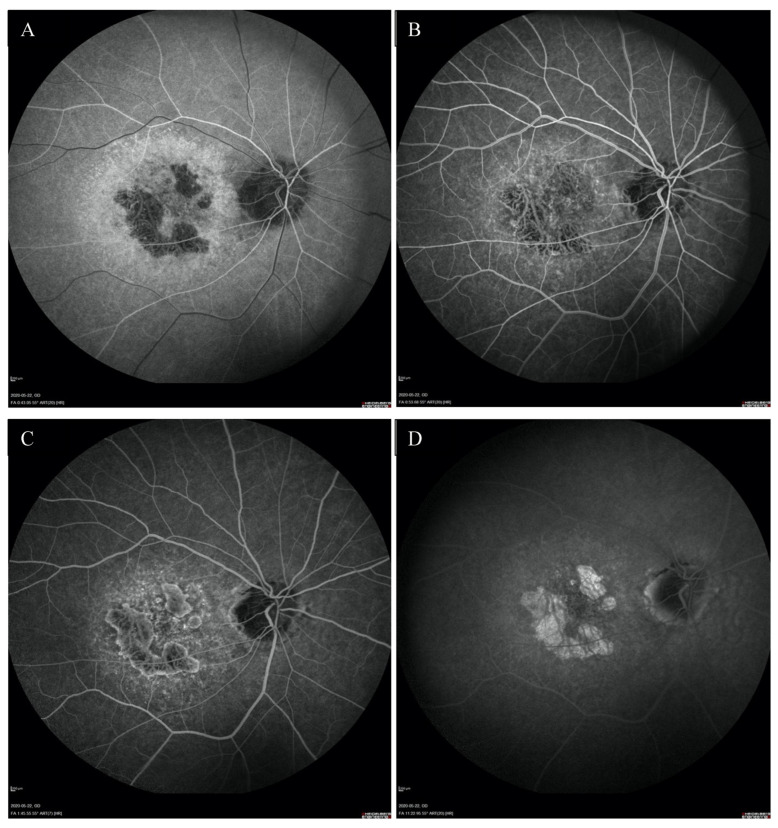
FFA of the right eye (**A**–**D**) revealed the presence of granular hyperfluorescence in the macular and concomitant areas of capillary atrophy.

**Figure 5 medicina-59-00399-f005:**
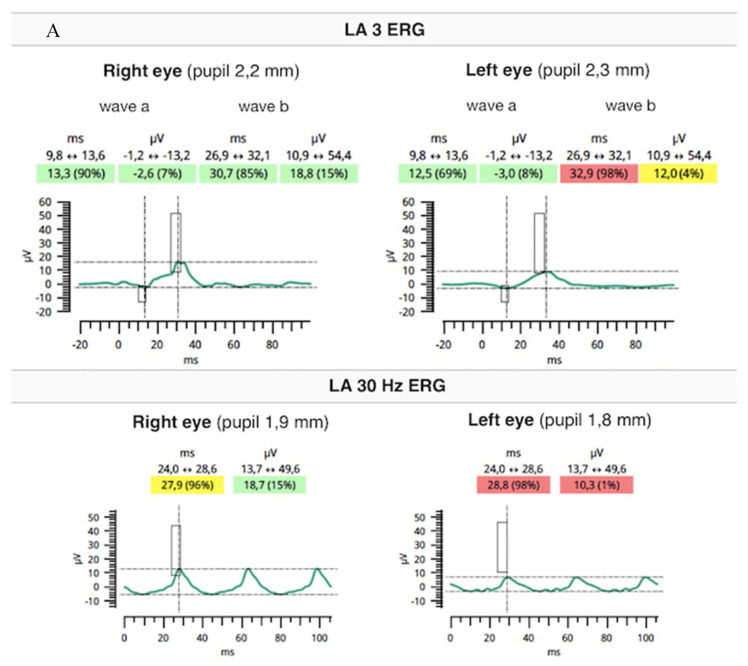
Flash ffERG showed lowering of a and b waves in light-adapted (LA) 3 ERG and LA 30 Hz ERG as well as prolonged peak times of a and b waves (significantly worse in the RE) (**A**). Dark-adapted (DA) 0.01 ERG and DA3 were within normal limits (**B**).

**Figure 6 medicina-59-00399-f006:**
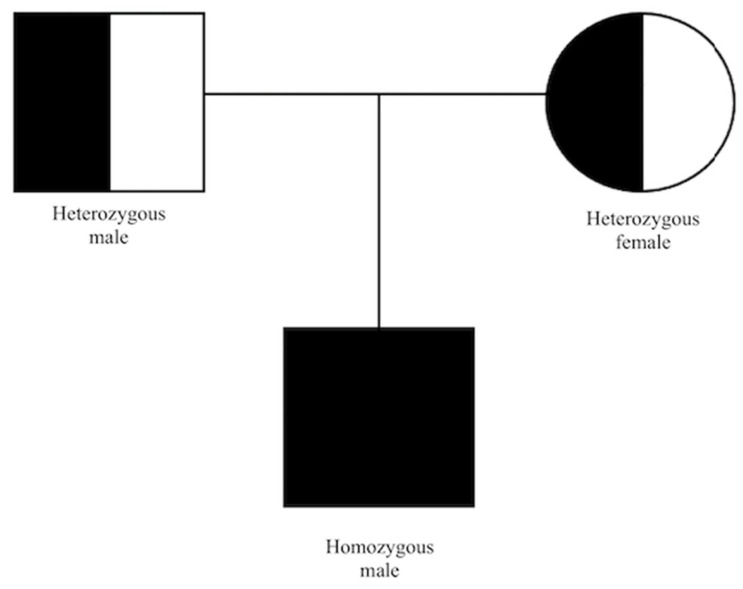
Pedigree chart of our patient.

## Data Availability

Not applicable.

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
