# Peer review of "CDHR1-Related Cone–Rod Dystrophy: Clinical Characteristics, Imaging Findings, and Genetic Test Results—A Case Report"

_medicina, 2023, doi:10.3390/medicina59020399_

Round 1

Reviewer 1 Report

The manuscript is well-written and actual. I have two recommendations: MfERG can be done to distinguish the diffuse from the local cone affection as the dark ERG is normal. It can help to distinguish electrophysiologically CRD from hereditary maculopathy. Nothing is mentioned about the parents, who should be heterozygotes.   

Author Response

We sincerely thank you for taking the time to review our manuscript and for your valuable comments.

We have highlighted all modifications made in the main text in red and added pedigree of our patient.

Reviewer 2 Report

The authors reported on a case report: CDHR1-related cone-rod dystrophy.

The clinical and molecular data very are very interesting and could be of interest to the readers. However, there are some comments that need to be addressed.

General comments:

+ Follow nomenclature: https://varnomen.hgvs.org/

*- The reference sequence (NP_) and NM_ should also be added.

**Gene name should be in italics. Kindly check.

USE (OMIM #), and use the proper name of the disorder.

Methods/Results

**IRB approval number should be mentioned.

Draw pedifree, Next Generation Sequencing (NGS)???

WES or WGS? Methodology??

Filtration steps?

Any other variants identified?

Was Sanger segregation performed? Add electrograms.

Results

It would be better to add the age, height, weight, and HC of the patients with SD.

How the variants were classified as pathogenic ACMG criteria fulfilled?

**Was the patient treated with any medication? Condition of patients before and after treatment?

Divide the Results part into headings, so the readers could easily understand each part.

Discussion

Any genotype-phenotype correlation? Location of variants associated with variable phenotypes?

In the last discussion add lines for future perspectives; discuss newborn screening, NIPT, PGT-A , PGT-M, for example.: Proper genetic counseling for the affected family is essential in the case of rare genetic diseases. Furthermore, parenteral genetic screening/diagnosis is the best strategy for managing this disease, which currently has no therapy (Alfadhel et al., 2019; Alyafee, Al Tuwaijri, et al., 2021; Alyafee, Alam, et al., 2021). Reporting additional cases associated with this gene would help identify genotype–phenotype correlations and lead to clinical trials in the future (Alfadhel et al., 2021).

Alfadhel, M., Umair, M., Almuzzaini, B., Alsaif, S., AlMohaimeed, S. A., Almashary, M. A., Alharbi, W., Alayyar, L., Alasiri, A., Ballow, M., AlAbdulrahman, A., Alaujan, M., Nashabat, M., Al-Odaib, A., Altwaijri, W., al-Rumayyan, A., Alrifai, M. T., Alfares, A., AlBalwi, M., & Tabarki, B. (2019). Targeted SLC19A3 gene sequencing of 3000 Saudi newborn: A pilot study toward newborn screening. Annals of Clinical Translational Neurology, 6(10), 2097– 2103. https://doi.org/10.1002/acn3.50898

Alyafee, Y., Al Tuwaijri, A., Alam, Q., Umair, M., Haddad, S., Alharbi, M., Ballow, M., Al Drees, M., Al Abdulrahman, A., Al Khaldi, A., & Alfadhel, M. (2021). Next generation sequencing based non-invasive prenatal testing (NIPT): First report from Saudi Arabia. Frontiers in Genetics, 12, 630787.

Alyafee, Y., Alam, Q., Altuwaijri, A., Umair, M., Haddad, S., Alharbi, M., Alrabiah, H., Al-Ghuraibi, M., Al-Showaier, S., & Alfadhel, M. (2021). Next-generation sequencing-based pre-implantation genetic testing for aneuploidy (PGT-A): First report from Saudi Arabia. Genes (Basel), 12(4), 461. https://doi.org/10.3390/genes12040461.

Alyafee Y, Al Tuwaijri A, Umair M, Alharbi M, Haddad S, Ballow M, Alayyar L, Alam Q, et al. Non-invasive prenatal testing for autosomal recessive disorders: A new promising approach. Front Genet. 2022 Nov 3;13:1047474. doi: 10.3389/fgene.2022.1047474.

Author Response

We sincerely thank you for taking the time to review our manuscript and for your valuable comments.

We have highlighted all modifications made in the main text in red and added pedigree of our patient. However, we do not have access to electrograms or information about filtration steps.

We do not have IRB approval number. It is a retrospective case report. All examinations complied with the requirements for good medical practice. A written informed consent was obtained from patients.
